# Cytoplasmic FBXO38 mediates PD-1 degradation

Xiwei Liu[1,12], Xiangbo Meng [2,12], Zuomiao Lin[3], Shutan Jiang[1], Haifeng Liu[1], Shao-cong Sun [4], Xiaolong Liu[1], Penghui Zhou [5], Xiaowu Huang[6,7,8], Lai Wei[9], Wei Yang[10,11] & Chenqi Xu [1,3✉]

Comment on: N Dibus et al (September 2024)

As a crucial molecule in T-cell immunity, PD-1 undergoes complex transcriptional and post-translational regulations (He and Xu, 2020). Our previous study demonstrated that FBXO38 mediates K48 ubiquitination of internalized PD-1, influencing T cell antitumor immunity (Meng et al, 2018). Our experimental systems included biochemical assays in both Jurkat T cells and HEK293 cells, a genetic model of conditional *Fbxo38* knockout specifically in T cells, and multiple tumor models in which FBXO38's role was assessed in T-cell exhaustion and immunotherapy. However, a recent study by Dibus et al, challenges our conclusion, considering that FBXO38 is dispensable for PD-1 regulation, based on experiments in a non-T cell system and a germline knockout model with severe developmental defects. They also evaluated FBXO38's role in an acute virus infection model, assessing only one late time point when T-cell responses had returned to baseline. We value these arguments and consider this correspondence as an opportunity to systematically review existing literatures and discuss discrepancy. Moreover, we performed new biochemical and cellular experiments in T cells to further clarify the regulations of PD-1 by FBXO38.

## Cytoplasmic localization of FBXO38

A central argument presented by Dibus et al, is that FBXO38, being restricted to the nucleus, cannot interact with the membrane protein PD-1, based on data from HEK293 cells with ectopically expressed PD-1 (Fig. 1 of Dibus et al, 2024). However, earlier studies have reported cytoplasmic localization of FBXO38 (Georges et al, 2019; Smaldone et al, 2004; Smaldone and Ramirez, 2006; Sumner et al, 2013) and identified a cytoplasmic substrate for this protein (Tian et al, 2023). FBXO38 contains one nuclear localization signal (NLS) and three nuclear export signals (NES), which facilitate its shuttling between the nucleus and cytoplasm (Smaldone and Ramirez, 2006) (Fig. 1A). Dibus et al, mentioned only the NLS, but the NES functions have been validated and should also be considered. When ectopically expressed in various cell lines, FBXO38 has been shown to localize in both the cytoplasm and nucleus (Fig. 7A of (Georges et al, 2019), Fig. 10 of (Smaldone et al, 2004), Fig. 1–5 of (Smaldone and Ramirez, 2006), Fig. 2 of (Sumner et al, 2013)). In some cells, cytoplasmic localization is predominant, while nuclear localization is minimal. Although overexpression might lead to the non-physiological accumulation of a nuclear protein in the cytoplasm, it is unlikely to render the nuclear signal nearly undetectable. A more plausible explanation is that FBXO38 undergoes post-translational modifications affecting nucleus/cytoplasm shuttling, similar to other NLS/NES-containing proteins. In the Dibus et al, study, the authors used a polyclonal FBXO38 antibody from Atlas Antibodies (HPA041444) to stain endogenous FBXO38 in HEK293 cells, concluding that it is restricted to the nucleus. However, the manufacturer's website indicates that this antibody stains FBXO38 in both the cytoplasm and nucleus. In an earlier publication by the same group, FBXO38 was observed in both soluble and insoluble fractions of cell lysis, while histone 3 was only in the insoluble fraction (Fig. 1B of (Dibus et al, 2022)). These inconsistencies suggest that cell culture conditions might affect FBXO38 localization. FBXO38 has three reported transcript isoforms: the full-length isoform 1, the Δ810-884 isoform 2 and the Δ640-884 isoform 3, migrating at 120-180 KD on gels (Fig. 1A). We utilized two antibodies: a HA-tag antibody (3F10) for *FBXO38-HA* knockin Jurkat cells, and a FBXO38 antibody (ab87729) for wild-type Jurkat cells (Fig. 1C–F). 3F10 detected all three FBXO38 isoforms, while ab87729 mainly detected isoform 1, likely because its epitope is missing in the shorter isoforms. Nevertheless, both antibodies were specific, as demonstrated in Fig. 1C (WT vs *FBXO38-KI*) and Fig. 1E (WT vs *FBXO38-KO*). We performed nucleus-cytoplasm fractionation and observed substantial FBXO38 in the cytoplasm, in either resting or stimulated T cells (Fig. 1D–F). Confocal imaging also supports the cytoplasmic localization of FBXO38 (Fig. 1G). Notably, a substantial fraction of endogenous PD-1 is also found in intracellular compartments, especially at endosomes (Fig. 1H). A proximity ligation assay demonstrated a direct physical interaction between FBXO38 and PD-1 in the cytoplasm (Fig. 1I), consistent with the

[1]CAS Center for Excellence in Molecular Cell Science, Shanghai Institute of Biochemistry and Cell Biology, Chinese Academy of Sciences, Shanghai, China. [2]Advanced Medical Research Institute, Meili Lake Translational Research Park, Cheeloo College of Medicine, Shandong University, Jinan, Shandong, China. [3]School of Life Science, Hangzhou Institute for Advanced Study, University of Chinese Academy of Sciences, Hangzhou, China. [4]Institute of Immunology, Chinese Institutes for Medical Research, Beijing, China. [5]State Key Laboratory of Oncology in South China, Collaborative Innovation Center for Cancer Medicine, Sun Yat-sen University Cancer Center, Guangzhou, China. [6]Department of Liver Surgery and Transplantation, Liver Cancer Institute, Zhongshan Hospital, Fudan University, Shanghai 200032, China. [7]Key Laboratory of Carcinogenesis and Cancer Invasion (Fudan University), Ministry of Education, Shanghai 200032, China. [8]Shanghai Key Laboratory of Organ Transplantation, Zhongshan Hospital, Fudan University, Shanghai 200032, China. [9]Guangdong Provincial Key Laboratory of Allergy & Clinical Immunology, The Second Affiliated Hospital, Guangzhou Medical University, Guangzhou 510000, China. [10]Guangdong Provincial Key Laboratory of Molecular Oncologic Pathology, Department of Pathology, School of Basic Medical Sciences, Southern Medical University, Guangzhou, China. [11]Department of Pathology, Nanfang Hospital, Southern Medical University, Guangzhou, China. [12]These authors contributed equally: Xiwei Liu, Xiangbo Meng. ✉E-mail: cqxu@sibcb.ac.cn
https://doi.org/10.1038/s44319-024-00254-y | Published online: 16 September 2024

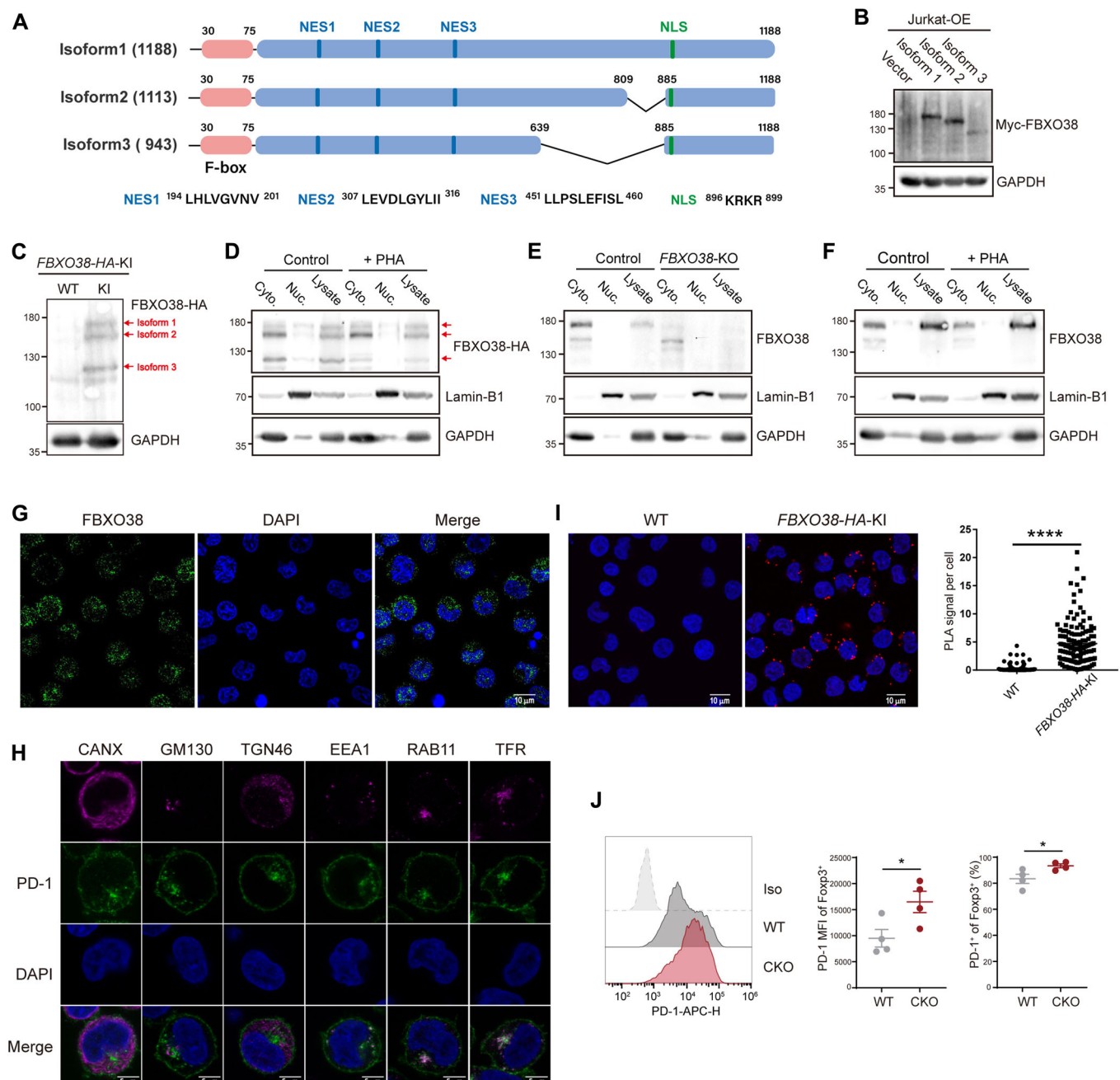

previous co-immunoprecipitation experiment in Jurkat T cells (Fig. 2e of (Meng et al, 2018)). The strict nuclear localization of FBXO38 in HEK293 cells used by Dibus et al, may explain the negative result of FBXO38-PD-1 co-immunoprecipitation in their study.

## PD-1 ubiquitination and degradation

Dibus et al, showed that the proteasomal inhibitor MG132 did not rescue PD-1 protein levels in HEK293 cells (Fig. 2c of Dibus et al, 2024). In contrast, MG132 treatment markedly enhanced PD-1 levels in HPB-ALL T cells (Fig. 3b of Dibus et al, 2024). The results from Jurkat T cells are inconsistent, while FACS showing a dramatic PD-1 increase upon MG132 treatment (Fig. 3c of Dibus et al, 2024), accompanying bar graph showed negligible effect (Fig. 3a of Dibus et al, 2024). These discrepancies suggest potential experimental issues. Nevertheless, three other groups

reported that MG132 treatment led to PD-1 accumulation in PBMC, Jurkat, HEK293, and RAW264.7 cells (Fig. 7E of (Lyle et al, 2019), Fig. 5G of (Zhang et al, 2020), Fig. 3A, C, S3A of (Zhou et al, 2020)), consistent with our findings (Fig. 1l–m of (Meng et al, 2018)). Since FBXO38 activity is dependent on neddylation, Dibus et al, treated cells with MLN4924 and observed a slight decrease in PD-1 levels in HEK293, Jurkat, and HPB-ALL cells (Fig. 2d, Fig. 3a–d of Dibus et al, 2024). Conversely, another

Figure 1. Intracellular localization of FBXO38 and the impact of *Fbxo38* knockout on PD-1 levels in Treg cells.

(A) The schematic representation of FBXO38 functional domains. The positions of the nuclear export signals (NES) and nuclear localization signal (NLS) are shown above. (B) Immunoblotting of the three FBXO38 splicing isoforms. Wild-type Jurkat cells were transduced with the constructs of three FBXO38 isoforms with a 3×Myc tag. FBXO38 was immunoblotted by anti-Myc. (C) Immunoblotting of endogenous FBXO38 in *FBXO38-HA* knockin Jurkat cells. Wildtype or *FBXO38-HA* knockin Jurkat cells were lysed and immunoblotted by anti-HA (3F10). The three isoforms are labeled by red arrows. (D–F) Nucleus-cytoplasm localization of endogenous FBXO38 in *FBXO38-HA* knockin (D) or wild-type (E, F) Jurkat cells. The total cell lysate, the cytoplasmic fraction, and the nuclear fraction of Jurkat were immunoblotted by anti-HA (3F10) or anti-FBXO38 (ab87729). Lamin-B1 and GAPDH were used as markers of the nuclear fraction and cytoplasmic fraction. *FBXO38-HA* knockin (D) or wildtype (F) Jurkat cells were either unstimulated or stimulated with PHA (150 ng/ml) for two days. In (E) wildtype or *FBXO38* knockout Jurkat cells were unstimulated. (G) Confocal imaging of endogenous FBXO38 in *FBXO38-HA* knockin Jurkat cells. FBXO38 were stained by anti-HA and subsequent secondary antibodies (α-rat-AF488). (H) Confocal imaging of endogenous PD-1 in wild-type Jurkat cells. Cells were stimulated with PHA (150 ng/ml) for 3 days to induce endogenous PD-1 expression. The stimulated cells were fixed, permeabilized, and stained with PD-1 and CANX (ER), GM130 (cis-Golgi), TRN46 (trans-Golgi), EEA1 (early endosome), RAB11, and TFR (recycling endosome) antibodies. Images were collected with a Leica SP8 confocal microscope. (I) Interaction of endogenous FBXO38 and PD-1 in *FBXO38-HA* knockin Jurkat cells, detected by the proximity ligation assay. The Left is the representative image, and the right is the PLA signal quantification. (J) PD-1 surface levels in wildtype and FBXO38-deficiency regulatory T (Treg) cells. Splenic Treg cells from WT and CKO mice at 18 months of age were gated by CD4+Foxp3+, and stained with anti-PD-1 (*n* = 4 mice). Iso indicates cells stained with isotype control antibody. Experiments c–i were independently repeated twice, while experiment j was repeated three times. (I, J) represent biological replicates. Error bars represent mean ± SEM. Statistical significance is indicated as *P < 0.05, ****P < 0.0001. Source data are available online for this figure.

group reported MLN4924 treatment upregulated PD-1 levels in Jurkat and HEK293 cells (Fig. 3C, S2C of (Zhou et al, 2020)). The reason for these conflicting MLN4924 results is unclear, but managing the cell toxicity of this drug is crucial to avoid misinterpretation.

We also assessed the effect of FBXO38 on PD-1 ubiquitination in T cells in our study. We found that knocking down or knocking out endogenous *FBXO38* decreased PD-1 ubiquitination levels and increased PD-1 protein levels in Jurkat cells (Fig. 2h,i, Fig. S2g–m of (Meng et al, 2018)). *FBXO38* overexpression had the opposite effect (Fig. 2g of (Meng et al, 2018)). These experiments were not included in the Dibus et al, study.

### FBXO38 functions in T-cell immunity

We generated *Fbxo38^flox/flox* mice and crossed them with *CD4^cre* mice to specifically delete FBXO38 in T cells. These *Fbxo38^CKO* mice had normal thymocyte development and peripheral homeostasis (Fig. S3 of (Meng et al, 2018)). Our results showed that FBXO38 deficiency increased PD-1 levels. Considering that PD-1 expression kinetics vary with stimulation strength (Fig. S5a,b of (Meng et al, 2018)), assessing a wide time window is crucial to fully understand FBXO38's role. In the tumor microenvironment, FBXO38 deficiency led to elevated PD-1 levels and impaired T-cell functions, resulting in accelerated tumor progression in melanoma and colorectal cancer models (Fig. 3d–l and Fig. S6 of (Meng et al, 2018)). Additionally, we conducted *Fbxo38* knockdown in mouse OT1 T cells to confirm its intrinsic role (Figs. S7, 8 of (Meng et al, 2018)). Our results are supported by other

studies showing that *FBXO38* knockdown or knockout upregulated PD-1 levels in Jurkat cells (Fig. 5F of (Zhang et al, 2020)) and human CAR-T cells (Fig. 4G of (Lv et al, 2023)). Moreover, IL8-mediated FBXO38 downregulation was associated with reduced PD-1 ubiquitination in CD8+ T cells (Fig. 5F-H of (Li et al, 2022)). To further strengthen our conclusion, we crossed the *Fbxo38^flox/flox* mice with the *Foxp3^YFP-cre* mice to specifically deplete FBXO38 in regulatory T (Treg) cells. In aged *Fbxo38^Treg-CKO* mice, FBXO38 deficiency significantly upregulated PD-1 levels (Fig. 1J).

Dibus et al, used a germline knockout strategy to deplete FBXO38 in all cells. Their previous study indicated that *Fbxo38*-KO mice exhibited severe developmental defects, including low birth rate, reduced body weight, and shortened body length (Dibus et al, 2022). To evaluate immune cell development, they checked splenocyte populations and found significant reductions in NK cells (Fig. EV1C of Dibus et al, 2024). Other immune populations, such as neutrophils and monocytes, showed differences, but high variability prevented reliable statistics. Their T cell subtype gating relied solely on CD62L, which should be paired with markers like CD44 for accuracy. Notably, FBXO38 is a defined coactivator of KLF7 that involved hematopoietic stem cell development (Schuettpelz et al, 2012; Smaldone et al, 2004). Thus, *Fbxo38* total knockout could affect early progenitors to skew T-cell immunity. To demonstrate minimal effects on T cell compartments, comprehensive analyses of thymocyte development and peripheral T cell functions, similar to our previous work, are necessary.

Using the *Fbxo38* total knockout model, Dibus et al, found that FBXO38 deficiency did not affect PD-1 levels in mouse CD4+ and CD8+ T cells under steady and activated states. However, they only assessed a single time point (Day 4) after activation, chosen based on published datasets of human T cells from PBMCs showing *PDCD1* transcription declines after Days 2–3. This reference is inappropriate, as human T cells differ significantly from naïve mouse T cells. Moreover, analyzing a single time point is insufficient. While naïve T cells express minimal levels of PD-1, Fig. 3H of Dibus et al showed substantial PD-1 expression in naïve cells, suggesting a need for improved staining protocols. Additionally, using B-cell depleted splenocytes as naïve T cells is imprecise, as effector and memory T cells are also present in the spleen.

Dibus et al, conducted acute LCMV Armstrong infection and assessed PD-1 levels on day 8, after viral clearance. Previous studies indicate that acute LCMV infection does not induce T cell exhaustion, with PD-1 levels returning to baseline postclearance (Barber et al, 2006). A more relevant model is a chronic LCMV C13 model in which exhausted T cells persistently express PD-1 (Barber et al, 2006), like the tumor models we studied. Again, assessing only one time point is insufficient to draw a solid conclusion. Moreover, they did not show any T cell functional data nor disease progression affected by FBXO38 deficiency. Fig. EV3 of Dibus et al has several caveats: (1) gating of PD-1+ cells did not cover the whole PD-1-expressing population; (2) *Fbxo38* HET data were missing in the left-side FACS panels; (3) FACS profiles in the upper-right panel showed slightly

higher PD-1 MFI in KO and HET T cells, contrasting with bar graph results. To draw convincing conclusions, the authors need to test FBXO38 deficiency in a disease model with sustained, high PD-1 expression and effective PD-1 blockade therapy. Moreover, a conditional knockout strategy is more physiologically relevant, for avoiding indirect effects from other cell types.

## Peer review information

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

## Acknowledgements

We thank Cell Biology facility of SIBCB and Animal Resource Center of Shandong University for technical support. All authors of Meng et al, 2018 have been informed of this correspondence and its content.

## Author contributions

**Xiwei Liu**: Data curation; Investigation; Writing—original draft; Writing—review and editing. **Xiangbo Meng**: Data curation; Investigation; Writing—original draft; Writing—review and editing. **Zuomiao Lin**: Investigation. **Shutan Jiang**: Writing—review and editing. **Haifeng Liu**: Writing—review and editing. **Shao-cong Sun**: Writing—review and editing. **Xiaolong Liu**: Writing—review and editing. **Penghui Zhou**: Writing—review and editing. **Xiaowu Huang**: Writing—review and editing. **Lai Wei**: Writing—review and editing. **Wei Yang**: Writing—review and editing. **Chenqi Xu**: Funding acquisition; Project administration; Writing—review and editing.

## Disclosure and competing interests statement

The authors have a patent filed for PD-1 ubiquitination.

