## [Peer Review File · EMBO Reports]

Cytoplasmic FBXO38 mediates PD-1 degradation

Chenqi Xu, Xiwei Liu, Xiangbo Meng, and Zuomiao Lin

Corresponding author(s): Chenqi Xu (cqxu@sibcb.ac.cn)

Review Timeline:

Submission Date:	15th Mar 24
Editorial Decision:	19th Jun 24
Revision Received:	2nd Aug 24
Editorial Decision:	13th Aug 24
Revision Received:	19th Aug 24
Accepted:	27th Aug 24

Transaction Report:

Dear Prof. Xu

Thank you for the submission of your Correspondence to Dibus et al and for your patience while it was reviewed. I am very sorry for the long delay but we have finally received a second report on it.

Both referees consider your line of argumentation convincing and valid. To proceed with the publication of your Correspondence as a response to Dibus et al, I kindly ask you to revise it according to the format requirements for Correspondences.

The article should fit on two printed pages, which corresponds to approx. 9,000 characters. Please do not include figures from published papers but refer to the figure panels in these papers instead. Please compile figures with your own data and include these together with the methods section in an Appendix. I suggest to answer the specific questions from the referees in the text, as no further experiments can be included at this stage, except for the experiment suggested by referee 1, which would help to clarify the issues further.

Please draft the Correspondence along these lines and I will, once you have resubmitted, see whether further changes are necessary.

I look forward to seeing a revised form of your manuscript when it is ready.

Kind regards,

Martina

Referee #1

Liu and colleagues response to the previously published observation by Dibus et al reflect on the different observations regarding FBXO38 function in PD-1 stability and abundance in T-cells. The correspondence presented here elaborates on the discrepancies in findings and tries to elude to the source thereof by detailed analysis of the model systems used and the conclusions made. As presented it is possible that the major source of 'disagreement' between both groups could arise from the models applied, however, the observations reported by Dibus et al as well by Liu et al are to be considered Valid within the timeframes and genetic models used.

The data presented supports the line of discussion by Liu and colleagues.

Regarding the presented data does the reviewer have questions:

Figure 1:

d) Could the authors please clarify why there are so many bands visible in unstimulated Jurkat cells when probed with the FBXO38 antibody? Are all bands present on the blot variants or isoforms, especially since they are absent in Figure 1b? Could the authors please provide an experiment where FBXO38 was silenced and show a western blot as presented in Figure 1d? Also, would it be possible to show an IF of Jurkat cells +/- PHA stimulation against endogenous FBXO38? This would significantly contribute to the discussion regarding nuclear to cytosolic FBXO38.

Referee #2

I find the response of Xu et al to be rational and compelling, pointing out all of the scientific and logical limitations associated with the Cermak et al manuscript. I would challenge Cermak with these issues before rendering a judgement on its acceptance, if their reviewers have not already done so.

I would say that the issue is not wholly resolved, as the FBXO38 may still be acting indirectly on its effect on PD1. If Xu and colleagues have not already done so, I would knock out the Ub acceptor lysine from the PD1 cytoplasmic domain, to show the ubiquitination dependency definitively (as had been done some years ago for MHC class II b chain, see Shin et al, 2006, Nature).

Xu should definitely be published if Cermak is.

I would correct "Novel Prize" to "Nobel Prize" in the introduction.

Referee #1: Liu and colleagues response to the previously published observation by Dibus et al reflect on the different observations regarding FBXO38 function in PD-1 stability and abundance in T-cells. The correspondence presented here elaborates on the discrepancies in findings and tries to elude to the source thereof by detailed analysis of the model systems used and the conclusions made. As presented it is possible that the major source of 'disagreement' between both groups could arise from the models applied, however, the observations reported by Dibus et al as well by Liu et al are to be considered Valid within the timeframes and genetic models used. The data presented supports the line of discussion by Liu and colleagues. Regarding the presented data does the reviewer have questions:

Figure 1:

d) Could the authors please clarify why there are so many bands visible in unstimulated Jurkat cells when probed with the FBXO38 antibody? Are all bands present on the blot variants or isoforms, especially since they are absent in Figure 1b? Could the authors please provide an experiment where FBXO38 was silenced and show a western blot as presented in Figure 1d? Also, would it be possible to show an IF of Jurkat cells +/- PHA stimulation against endogenous FBXO38? This would significantly contribute to the discussion regarding nuclear to cytosolic FBXO38.

We appreciate the reviewer for acknowledging the validity of our data, as well as for their thorough analysis and constructive suggestions.

This reviewer raised a concern about the specificity of FBXO38 antibody HPA034821 from Atlas Antibiotics. Indeed, the immunoblotting with this antibody showed multiple bands even in the FBXO38 knockout cells (previous Fig. 1c). We therefore used another FBXO38 antibody ab87729 from Abcam to detect endogenous FBXO38, which exhibits high specificity. Moreover, we used a well-validated HA-tag antibody 3F10 from Roche to detect endogenous FBXO38 in *FBXO38-HA* knockin Jurkat T cells. 3F10 detected all three FBXO38 isoforms (new Fig. 1c, d), while ab87729 mainly detected isoform 1 likely because its epitope is missing in the shorter isoforms (new Fig. 1e, f). Nevertheless, both antibodies were specific, as demonstrated in the new Fig. 1c (WT vs *FBXO38-KI*) and the new Fig. 1e (WT vs *FBXO38-KO*). We performed nucleus-cytoplasm fractionation and observed substantial FBXO38 in cytoplasm, in either resting or PHA-stimulated T cells (new Fig. 1d-f). These new data further validate our conclusion that FBXO38 is present in the cytoplasm of T cells.

The reviewer also suggested an immunofluorescence experiment on Jurkat cells with and without PHA stimulation to examine endogenous FBXO38 distribution. While we agree this would provide further evidence, our attempts using the FBXO38 antibodies, either the previous HPA034821 or the current ab87729, showed substantial noise signals even in FBXO38 knockout cells. Nevertheless, the immunofluorescence staining with 3F10 was specific, and validated the presence of FBXO38 in the cytoplasm (Fig. 1g).

In summary, both the immunoblotting and immunofluorescence experiments validate that a substantial fraction of FBXO38 is present in the cytoplasm of T cells, which is consistent with the existing literatures.

Figure 1. Intracellular localization of FBXO38 and the impact of *Fbxo38* knockout on PD-1 levels in Treg cells.

a, The schematic representation of FBXO38 functional domains. The positions of the nuclear export signals (NES) and nuclear localization signal (NLS) are shown above.

b, Immunoblotting of the three FBXO38 splicing isoforms. Wildtype Jurkat cells were transduced with the constructs of three FBXO38 isoforms with 3×Myc tag. FBXO38 was immunoblotted by anti-Myc.

c, Immunoblotting of endogenous FBXO38 in *FBXO38-HA* knockin Jurkat cells. Wildtype or *FBXO38-HA* knockin Jurkat cells were lysed and immunoblotted by anti-HA (3F10). The three isoforms are labeled by red arrows.

d-f, Nucleus-cytoplasm localization of endogenous FBXO38 in *FBXO38-HA* knockin (**d**) or wildtype (**e-f**) Jurkat cells. The total cell lysate, the cytoplasmic fraction, and

the nuclear fraction of Jurkat were immunoblotted by anti-HA (3F10) or anti-FBXO38 (ab87729). Lamin-B1 and GAPDH were used as markers of the nuclear fraction and the cytoplasmic fraction. *FBXO38-HA* knockin (**d**) or wildtype (**f**) Jurkat cells were either unstimulated or stimulated with PHA (150 ng/ml) for two days. In **e**, wildtype or *FBXO38* knockout Jurkat cells were unstimulated.

g, Confocal imaging of endogenous FBXO38 in *FBXO38-HA* knockin Jurkat cells. FBXO38 were stained by anti-HA and subsequent secondary antibody (α -rat-AF488).

h, Confocal imaging of endogenous PD-1 in wildtype Jurkat cells. Cells were stimulated with PHA (150 ng/ml) for three days to induce endogenous PD-1 expression. The stimulated cells were fixed, permeabilized and stained with PD-1 and CANX (ER), GM130 (cis-Golgi), TRN46 (trans-Golgi), EEA1 (Early Endosome), RAB11 and TFR (Recycling Endosome) antibodies. Images were collected by Leica SP8 confocal microscope.

i, Interaction of endogenous FBXO38 and PD-1 in *FBXO38-HA* knockin Jurkat cells, detected by the proximity ligation assay. Left is the representative image and right is the PLA signal quantification.

j, PD-1 surface levels in wildtype and *FBXO38*-deficiency regulatory T (Treg) cells. Splenic Treg cells from WT and CKO mice at 18-month age were gated by $CD4^+Foxp3^+$, and stained with anti-PD-1 (n=4). Iso indicates cells stained with isotype control antibody.

Data in panel i-j were analyzed by two-tailed unpaired *t*-test (**i** right, **j** right). Error bars represent mean \pm SEM. * $P < 0.05$, **** $P < 0.0001$.

Referee #2: I find the response of Xu et al to be rational and compelling, pointing out all of the scientific and logical limitations associated with the Cermak et al manuscript. I would challenge Cermak with these issues before rendering a judgement on its acceptance, if their reviewers have not already done so.

I would say that the issue is not wholly resolved, as the FBOX38 may still be acting indirectly on its effect on PD1. If Xu and colleagues have not already done so, I would knock out the Ub acceptor lysine from the PD1 cytoplasmic domain, to show the ubiquitination dependency definitively (as had been done some years ago for MHC class II b chain, see Shin et al, 2006, Nature).

Xu should definitely be published if Cermak is.

I would correct "Novel Prize" to "Nobel Prize" in the introduction.

We thank the reviewer for considering our work rational and compelling, as well as for their careful analysis and constructive suggestions.

This reviewer suggested performing an experiment to mutate the lysine ubiquitination sites of PD-1, similar to the approach used in a previous study on MHC II. We have already conducted the K-to-R mutation experiment as suggested, which is detailed in

our prior publication (Meng et al., Nature, 2018, Figures 2F, 2K, 2O). Our results showed that mutating lysine residues in the PD-1 cytoplasmic domain significantly reduced ubiquitination mediated by FBXO38 and stabilized protein levels, thereby conclusively demonstrating the dependency on ubiquitination.

Thank you for pointing out the spelling error. This part of introduction has been removed from the revised version due to word limit.

REDACTED: Figure 2 of Meng *et al.*, Nature 2018

Dear Chenqi,

Thank you for the submission of your revised Correspondence. I have now discussed it with our Chief Editor and we both agree that the text would need some minor revision before we could accept it for publication.

- After discussing the manuscript structure within the team, we decided to move your data figure to the main text. Please provide the figure as high-resolution figure file and refer to the figure as Figure 1.
- The Methods will remain part of the Appendix.
- Please fill our Author Checklist, which you can download from our Guide to Authors.
- You currently have 21 references, which should be reduced as much as possible for a Correspondence. Please see my suggestions in the attached manuscript file.
- Please provide source data for Figure 1. We need the minimally modified data (Western blots, imaging data, quantification data). Please make a folder called 'Figure 1' with subfolders for each figure panel and upload this as zipped file.
- Please provide up to 5 keywords.
- Data availability section: Please simply state that "No data have been deposited in public databases".
- References need to be alphabetical, et al should be used after 10 author names. DOIs should only be used for preprints and datasets that have not been published yet. The year should be in brackets.
- Please add information on funding in the Acknowledgments and in the online manuscript tracking system, if relevant.
- Moreover, we note that not all authors of Meng et al, Nature 2018, are contributing authors on the Correspondence. Please let all authors of the former manuscript know about this Correspondence and offer them to sign on as authors with contact@emboreports.org in CC. It is not mandatory that all former authors sign in but they should be informed.

Please return the revised manuscript within one week.

Please inform us about authorship changes by September 9th. Your Correspondence will be typeset meanwhile and we can update authorship at proof stage.

Dibus et al and your Correspondence will be co-released a few days after September 9th, unless all authorship changes have been completed before that date, in which case both manuscripts will be co-published once typesetting is complete.

Please contact me any time of you have any questions. I look forward to seeing a revised form of your manuscript when it is ready.

Kind regards,

Martina

The authors have addressed all minor editorial requests.

Prof. Chenqi Xu

CAS Center for Excellence in Molecular Cell Science, Shanghai Institute of Biochemistry and Cell Biology; University of Chinese Academy of Sciences, Chinese Academy of Sciences

320 Yue Yang Road

Shanghai 200031

China

Dear Chenqi,

Thank you for the submission of your revised Correspondence to our editorial offices. I have now accepted it for publication.

Your manuscript will be copy edited and you will receive page proofs prior to publication. Please note that you will be contacted by Springer Nature Author Services to complete licensing information.

Please contact me any time if you have any questions.

Kind regards,

Martina

Martina Rembold, PhD

Senior Editor

EMBO reports